# Evolution and Current Challenges of Sustainable Consumption and Production

Peter Glavič 

Department of Chemistry and Chemical Engineering, University of Maribor, Smetanova 17, 2000 Maribor, Slovenia; peter.glavic@um.si

**Abstract:** This review paper examines the past, present, and future of sustainable consumption and production (SCP). The history of the Sustainable Development Goal No. 12 (i.e., to ensure sustainable consumption and production patterns) is presented and analyzed. A definition of the sustainable consumption is given and the role of education is explained. The present status and existing trends of SCP are introduced by analyzing unsustainable behavior and the existing dilemma, namely sustainable growth or degrowth. A very broad range of methods is used for measuring and evaluating SCP within sustainable development. To forecast the future of SCP, important trends are presented. The future development of SCP will follow several megatrends and it will require reduced personal and collective consumption (degrowth). Energy usage in buildings, renewable energy sources, and energy storage will be important in that respect. Transportation emissions will continue to be lowered. Waste, especially food waste, shall be reduced, and consumer products shall become more durable. All waste must be collected and separated to be reused. SPC is elaborated in view of the two approaches—Industry 4.0 (smart factory), and the "Sixth Wave" evolution. Net-zero greenhouse gas emissions, resource efficiency, and zero waste will be at the forefront of future activities. A circular economy requires extension of product lifetimes, and the reuse and recycling of products. Reducing emissions, pollution and specific energy, water, and raw material usage (especially critical raw materials), as well as the role of digitalization, will be important.

**Keywords:** sustainable consumption; sustainable production; evolution; policy; education



## 1. Introduction

Three decades after the Rio conference, several United Nations (UN), European Union (EU) bodies, and nongovernmental organizations (NGOs) have expressed their anxiety regarding the results achieved with respect to sustainable development (SD) in general and SCP in particular. According to the GlobeScan-SustainAbility survey [1], experts are critical about achievements in Global Goal 12 in society. Only 9% rate it as good (4 + 5), but 59% as poor (1 + 2) while it ranks 2nd within the most urgent goals (after climate action). The progress since 2017 is below the average (10th out of the 17 SDGs) and is ranked 12th based on mean score.

### 1.1. Recent Data about the Need for Action

As shown in the European Environment Agency's (EEA's) five-yearly flagship report [2], "Europe needs to find ways to transform the key societal systems that drive environment and climate pressures and health impacts—rethinking not just technologies and production processes but also consumption patterns and ways of living." The outlook to 2030 suggests that the current rate of progress will not be sufficient to meet 2030 and 2050 climate and energy targets. Europe needs a new growth strategy that will transform the EU into a modern, resource-efficient, and competitive economy [3]. The EU aims to be climate-neutral by 2050 with net-zero greenhouse gas (GHG) emissions. The European Green Deal action plan requires efficient use of resources by moving to a clean, circular economy.

The resulting climate change is bringing elevated temperatures with floods and avalanches, storms, hurricanes, draughts, fires, the Arctic ice and glaciers melting, sea levels rising, species' extinction, invasive species spreading out, and new diseases [4]. The temperature change due to GHG emissions is following the 'hysteresis' effect; the temperature was growing slowly with the increasing GHG concentration in atmosphere at the beginning and is speeding up exponentially later to approach a new equilibrium state [5]. If we wanted to lower the temperatures to preindustrial levels, the hysteresis curve would require the GHGs to be lowered under the concentrations in the preindustrial era; by lowering them to the preindustrial level only, the temperatures will not lower to the expected level.

Besides the atmospheric GHG ($CO_2$, $CH_4$, $NO_x$, etc.) emissions causing climate change, civilization has crossed additional three of nine 'planetary boundaries': extinction rate (biosphere integrity), deforestation (land-system change), and flow of nitrogen and phosphorus (biogeochemical flows) [6]. The boundary for ozone depletion is an example of how political action requiring banning chlorofluorocarbons (CFCs) released the pressure in one of the boundaries. The other four boundaries are as follows: ocean acidification, freshwater use, atmospheric aerosol loading, and chemical pollution with radioactive and nanomaterials (introduction of novel entities). The Doughnut economics (regenerative and distributive economy) approach has the nine planetary boundaries outside the doughnut, and the twelve dimensions of social foundation (food, water, energy, housing, health, education, income and work, peace and justice, political voice, social equity, gender equality, and networks) inside of it [7]. Between the two layers, a safe and just space for humanity exists; it does not overshoot the planetary boundaries and is not short of falling away from a solid social foundation.

Less than 3% of the world's water is fresh (drinkable); since 2.5% of it is frozen in the Antarctic and Arctic cover and in glaciers, humanity must rely on 0.5% of global water for all of human ecosystem and needs. Water is being polluted faster than nature can recycle and purify it in rivers and lakes [8]. Only 17.5% of the world's final energy consumption in 2016 was from renewables [9]. Substantial environmental impacts from food occur in the production phase (agriculture, food processing). Humans' dietary choices and habits affect the environment through food-related energy consumption and waste generation. Land degradation, declining soil fertility, unsustainable water use, overfishing, and marine environment degradation are all diminishing the ability of our natural resource base to supply food.

We are far from the SDGs in the other two constituent parts of SCP, too. In 2016, 9.6% of the workforce aged 18–64 in the EU countries was "working poor" (i.e., having less than 60 % of average income [10]); 21.1% of the population in the EU (92.4 million people) were at risk of poverty or social exclusion in 2019 [11]. In 2019, the EU gender gap in the employment rate was 11.7%, and in hourly gross earnings it was 14.4%. Only 26 individuals own the same wealth as the 3.7 billion people who make up the poorest half of humanity globally.

The developing countries in the EU (EU-13) received only 4.4% of research and innovation funding in Horizon 2020 while having 20.6% of the EU's population. Therefore, their researchers, medical doctors, and entrepreneurs are moving to developed countries, thereby increasing the overall costs and non-equalities in developing countries. We have to reverse these trends to meet our Europe 2020 goals and make the following United Nations Development Strategies true: "Health, demographic change and wellbeing; inclusive, innovative and reflective societies; freedom and security of world citizens."

Should the global population reach 9.6 billion by 2050, the equivalent of almost three planets could be required to provide the natural resources needed to sustain current lifestyles [12]. Close to 900 million people still do not have access to drinking water. While almost 690 million people go undernourished and another 820 million hungry, 1.0 Gt (billion tones) of food is wasted every year. On the other side, 1.9 billion people globally are

overweight or obese what is detrimental to their health and the environment. "68.5 million people had been driven from their homes across the world at the end of 2017" [13].

The present consumption trends are economically unviable [12]. Despite technological advances that have promoted energy efficiency gains, energy use in OECD countries was supposed to grow another 35% by 2020. In 2020, the COVID-19 pandemic has temporarily slowed down the increase. Commercial and residential energy use is the second most rapidly growing area of global energy use after transport. In 2002 the motor vehicle stock in OECD countries was 550 million vehicles (the global number is estimated to 1.4 billion cars, trucks, and buses), 75% of which were personal cars. A 32% increase in vehicle ownership was expected by 2020. At the same time, motor vehicle road distances were expected to increase by 40%, and global air travel before the COVID-19 crisis was projected to triple in the same period. Households consume 29% of global energy and consequently contribute to 21% of resulting GHG emissions. The food sector consumes about 30% of total energy in the world and emits around 22% of total GHGs.

*1.2. Methodology and Scope*

This review paper built on the additional literature searches, proceedings of the last SCP conferences, and the author's personal experiences. The literature basis consisted of documents and reports from UN, OECD, EU, national and international associations and academies, periodic reports of environmental, social, and economic organizations, scientific papers, and World Wide Web searches.

Besides the historical background, the paper aims to examine the present status and challenges of the SCP. Measuring SCP is developing, but there are many problems connected with it. An example from the European chemical industry shows a good practice. UNIDO, the United Nations Industrial Development Organization is developing the concept of Green Industry for different sectors. USA and Chinese academies of engineering have discussed the present grand challenges.

Future development of SCP requires a more sophisticated approach. Global megatrends and long-term development mechanisms will be important in this respect. Two scenarios will be discussed, the Industry 4.0 concept and the wavelike human development with its "6th wave" of innovation. The resulting future trends in SCP will be shown separately for the sustainable consumption (SC) and sustainable production (SP). They will enable the drawing of some general conclusions.

## 2. Sustainable Consumption and Production History

*2.1. The Start and Development of Sustainable Consumption and Production*

"Greening of industry" started in the 1960s. Environmental protection using end-of-pipe treatment of emissions and reduced concentration of pollutants was introduced in the 1970s [14]. In the 1980s the focus changed to pollution prevention and cleaner technologies. Environmental management and cleaner products became active in the early 1990s. The SCP term has been introduced at the United Nations Conference on Environment and Development (UNCED) in Rio de Janeiro, 1992 [15]. Chapter 4 of Agenda 21 contained two areas of endeavor:

(a)    The existing ways of production and consumption are unsustainable, and
(b)    National policies and strategies shall foster changes in these ways.

Agenda 21 argued that "the major cause of the continued deterioration of the global environment are the unsustainable patterns of consumption and production, particularly in industrialized countries, which is a matter of grave concern, aggravating poverty and imbalances" (ibid, 4.3). "Action is needed to promote patterns of consumption and production that reduce environmental stress and will meet the basic needs of humanity."

The SCP idea was discussed in the Johannesburg Plan of Implementation [16] during the World Summit on Sustainable Development in 2002 (WSSD). SCP was recognized as one of the three most important requirements for future sustainable development: "Fundamental changes in the way societies produce and consume are indispensable for achieving

global sustainable development with the developed countries taking the lead and with all countries benefiting from the process". "Poverty eradication, changing unsustainable patterns of production and consumption, and protecting and managing the natural resource base of economic and social development are essential requirements" (ibid, paragraph 2).

The academic community quickly picked up the SCP concept after the Johannesburg conference. Many projects have been carried out in the period 2005–2010. Tukker et al. [17] edited a book on radical changes to SCP, and Lahlou [18] edited case studies in SCP. Tukker coauthored a paper on fostering the change to SCP [19]. Medina wrote about scavengers, salvagers for SCP [20]; about 1% of the urban population (15 million people worldwide) survives by reusing products, clothes, and food from municipal waste.

The Rio+20 summit in 2012 adopted the objectives of the 10-Year Framework of Programs (10YFP) on SCP [21] proposed to "accelerate the shift towards SCP, improve resource efficiency, decouple economic growth from environmental degradation and resource use, and create decent jobs". They were mentioning awareness raising, sharing information and knowledge, capacity building, financial and technical assistance for developing countries, etc.

In 2014, the United Nations Environment Program (UNEP) [22] declared SCP a "holistic approach about systemic change", built around three main objectives:

- "Decoupling environmental degradation from economic growth by doing more and better with less" (more goods and services with less impact—environmental degradation, waste, and pollution).
- "Applying the lifecycle thinking" to increase sustainable management of resources, improve resource efficiency in production and consumption phases of the lifecycle from resource extraction, to use and re-use products and services before waste disposal.
- "Sizing opportunities for developing countries" (SCP contributes to the achievement of the sustainable development by creating "new markets, green and decent jobs and more efficient, welfare-generating natural resource management in developing countries". It is based on environmentally sound and competitive technologies).

Today, "SCP is about promoting resource and energy efficiency, sustainable infrastructure, and providing access to basic services, green and decent jobs, and a better quality of life for all. Its implementation helps to achieve overall development plans, reduce future economic, environmental and social costs, strengthen economic competitiveness, and reduce poverty" [12].

### 2.2. Resource Efficient and Cleaner Production Branch of SCP

After the Rio Conference, UNIDO and UNEP commenced some pilot projects in selected countries to develop preventive environmental activities. In 1994 they started with the National Cleaner Production Centers (NCPCs). Later, UNIDO and UNEP have included resource efficiency agenda besides the cleaner production one. The Global Network for Resource Efficient and Cleaner Production (RECP net) has been developed. It is now joining 58 developing and transition countries [23]. In 2017, the 65 members accepted the Nairobi Declaration [24] in which they stressed the importance of all the three domains of sustainable development:

1. Economic domain: manufacturing and service companies which produce income and jobs while increasing efficient use of natural resources (raw-materials, water, and energy).
2. Social domain: to reduce poverty, and improve well-being while respecting natural resource scarcities, and ensuring safe and healthy production.
3. Environmental domain: to ensure low carbon, resource efficient and green industrialization using environmental management, and reducing emissions.

The partner countries promised to intensify cooperation on national, regional, and global levels by creating knowledge and techniques and sharing best practices. They were determined to achieve green industry and green growth by applying RECP policies,

methods, and techniques, especially among small and medium-sized enterprises (SMEs). Nowadays, the NCPCs are raising awareness on RECP; they demonstrate its benefits in the three SD pillars, help to finance RECP investments, and disseminate technical information [25].

### 2.3. Roundtables for SCP

Following the Rio conference, regional Roundtables for Cleaner Production (RCP) have been initiated by UNEP and UNIDO. After the Johannesburg World Summit on Sustainable Development, they have been transformed into Roundtables for Sustainable Consumption and Production (RSCP). The first European Roundtables for Cleaner Production (ERCP) took place in 1994 in Graz, Austria. Since the 9th conference in Bilbao (2004) they were organized under the new name European Roundtable for Sustainable Consumption and Production (ERSCP). The last ERSCPs have been attended by 140–450 participants each, and the 20th meeting is organized in Graz, Austria, again.

ERSCP conferences have concentrated on production and consumption in industry. In 2012, the Sustainable Consumption Research and Action Initiative network (SCORAI Europe) has been founded to include elements of individual consumers' behavior. SCORAI members are "professionals working at the interface of material consumption, human well-being, and technological and cultural change". The 2013 ERSCP conference, co-organized by the Environmental Management for Sustainable Universities (EMSU) network, was beneficial for the ERSCP conference participants too. Cooperation with the European Environment Information and Observation Network (EIONET) and closer contacts with the Asia Pacific RSCP and the African RSCP started. An interesting background to the ERSCP is that until 2013, there was no official organization responsible for hosting the ERSCPs. An informal group of experts decided each time where and when the next ERSCP would be hosted. This changed when a formal 'ERSCP Society' under Dutch law was set up, as a membership organization which democratically elects its board, and where this board has the task to foster SCP knowledge in Europe, also via the ERSCP [26].

The North American Roundtable on SCP [27] was established in January 2010 after the North American Sustainable Consumption Alliance (NASCA) had joined with the Citizens Network for Sustainable Development. The new SCORAI US, and the Canadian Environmental Network (CEN) were active since 2001. The partnership provided opportunities for further discussion about SCP, connected with the talks on the "green economy" but they held no conferences. A list of key recommendations emerged from the various workshops linked to SCP over the years 2001–2013.

The Asia Pacific Roundtable on Sustainable Consumption and Production (APRSCP) was established in 1997, and it had its 14th event in 2018. The supporting partners were UNEP, UNIDO, Deutsche Gesellschaft für Internationale Zusammenarbeit (GIZ), and the European Union (EU SWITCH-Asia Policy Support Component, PSC). The African Roundtable on Sustainable Consumption and Production [28] is a regional nongovernmental, not for profit organization with a mission of promoting SCP in Africa. It was established by the African SCP practitioners in Casablanca in 2004 and organized its 9th event in 2016. The South American Roundtable for SCP has not been perceived, so far.

The UK Institution of Chemical Engineers is publishing a journal entitled Sustainable Production and Consumption. It is an official journal of the European Federation of Chemical Engineering.

### 2.4. The Way to Sustainable Development Goals

Eight Millennium Development Goals (MDGs) [29] were announced in 2008 and started in 2010. They included Goal No. 7: "To ensure environmental sustainability". Although the SCP was not explicitly mentioned, "reversing the loss of environmental resources" was one of the targets. Two years later, the Johannesburg World Summit on Sustainable Development (WSSD) recognized the need for a 10YFP on SCP. The Johannesburg Plan of Implementation of the WSSD reaffirmed that "fundamental changes in the

way societies produce and consume are indispensable for achieving global sustainable development" [16]. This was followed by the initiation of the Marrakech Process in 2003, which "developed various mechanisms, including regional consultations, Task Forces, and dialogues with different stakeholders, in order to refine the concept of SCP and to show how it could be made operational in very different countries, economic sectors and cultural contexts" [30]. This work has provided a major input for the development of the 10YFP, which was formally accepted at the United Nations Conference on Sustainable Development (Rio+20) in June 2012.

The 10YFP is a global framework for action to enhance international cooperation and accelerate the shift towards SCP patterns in both developed and developing countries. The framework supports capacity building and facilitates access to technical and financial assistance for this shift in developing countries. It aims at developing, replicating, and scaling up SCP and resource efficiency initiatives at national and regional levels, decoupling environmental degradation and resource use from economic growth and thus increasing the net contribution of economic activities to resource efficiency and productivity, poverty eradication, social development, and environmental sustainability [22].

Rio+20 resulted in the political document "The future we want" [31] which contained, "clear and practical measures for implementing sustainable development." In Part I, "Our Common Vision," it claimed that "changing unsustainable and promoting sustainable patterns of consumption and production, protecting and managing the natural resource base of economic and social development are the overarching objectives of and essential requirements for sustainable development". The UN General Assembly endorsed the final document in September 2012.

At Rio+20, Member States also decided to start a process "to develop a set of Sustainable Development Goals (SDGs) which will build upon the MDGs". The Conference also "adopted guidelines on green economy policies". In Part III, "Green economy," they affirmed to promote SCP patterns. The document recognized that the need for "urgent action on unsustainable patterns of production and consumption . . . remains fundamental in addressing environmental sustainability and promoting conservation, sustainable use of biodiversity and ecosystems, regeneration of natural resources, and the promotion of sustained, inclusive, and equitable global growth."

In 2015, the "new UN 2030 Agenda for Sustainable Development with 17 Sustainable Development Goals and 169 targets was accepted" [32]. Goal No. 12 is to "ensure responsible consumption and production patterns". Addressing social and economic development within the carrying capacity of ecosystems and decoupling economic growth from environmental degradation is an essential requirement for sustainable development. Paragraph 28 of the Agenda reads: "We (Countries) commit to making fundamental changes in the way that our societies produce and consume goods and services". The Goal contains eleven targets ranging from managing natural resources, wastes and chemicals, to tasks of countries, companies, and the general public.

"Sustainable lifestyles and sustainable patterns of consumption and production" are also mentioned in the Paris Agreement, "the first-ever universal, legally binding global climate deal" adopted by 195 countries [33]. Its goal is to limit global warming "well below 2 °C"—trying to limit it to 1.5 °C. The latter would require zero GHGs emissions somewhere between 2030 and 2050.

The European Commission (EC) accepted the SCP and Sustainable Industrial Policy Action Plan [34]. It included "a series of proposals on SCP that should contribute to improving the environmental performance of products and increase the demand for more sustainable goods and production technologies". The EU made a positive and constructive contribution to the creation of the 2030 Agenda for Sustainable Development. A series of publications have been issued highlighting the work of the EC towards the aims set out in the SCP Action Plan. European SCP Policies are including energy and material resource efficient economy, circular economy, eco-innovation, eco-design, the Eco-Management and

Audit Scheme (EMAS), eco-labelling, green public procurement, integrated product policy, waste prevention and recycling, etc.

## 2.5. Sustainable Consumption

SCP has two constituents, consumption and production. Shall we treat them separately or together? Sustainable production is "the creation of goods and services using processes and systems that are non-polluting, conserving of energy and natural resources, economically viable, safe and healthful for workers, communities, and consumers" [35]. Typical activities include circular economy, cleaner production, pollution prevention, integrated pollution prevention and control (IPPC), best available techniques (BAT), responsible care, process optimization, energy integration, recycling, reuse, repair, regeneration, remanufacturing, renewable resources, factor X, eco-efficiency, industrial ecology, supply chain, life cycle assessment, doing more with less, environmental accounting, social responsibility, global reporting initiative, etc.

SC was defined at the Oslo Symposium in 1994 [36] as, "the use of services and related products, which respond to basic needs and bring a better quality of life while minimizing the use of natural resources and toxic materials as well as the emissions of waste and pollutants over the life cycle of the service or product so as not to jeopardize the needs of further generations".

SC is the use of products and services that have a minimal impact on the environment and enable future generations to meet their needs. It can be regarded on different levels including global, state, region, city, community, or enterprise, public institution, household or individual levels. The Paris Climate Agenda has been signed at the global level. Strategies, action plans, roadmaps, directives, and communications were accepted at the EU level while laws and other regulations are used at the state level. SC is mostly regarding materials (especially critical raw materials), energy, water, resource efficiency, and (zero) waste.

Individual consumption is connected to consumption patterns, life cycle thinking, and lifestyles (habits). The European Innovation Partnership on Smart Cities and Communities [37], "brings together cities, industry and citizens to improve urban life through more sustainable integrated solutions". Education for sustainable development (ESD) is to be mentioned in this context, too. Smart home, mobility, food consumption, and carbon footprint are typical keywords.

Sustainable lifestyles are important for future SC. "A sustainable lifestyle is a cluster of habits and patterns of behavior embedded in a society and facilitated by institutions, norms and infrastructures that frame individual choice, in order to minimize the use of natural resources and generation of wastes, while supporting fairness and prosperity for all" [38]. "A sustainable lifestyle minimizes ecological impacts while enabling a flourishing life for individuals, households, communities, and beyond. It is the product of individual and collective decisions about aspirations and about satisfying needs and adopting practices which are in turn conditioned, facilitated, and constrained by societal norms, political institutions, public policies, infrastructures, markets, and culture" [39].

The EU funded project Sustainable Consumption Research Exchanges [40] organized several workshops and conferences. Many reports, papers in special issues, and books have been published by them (see e.g., J. of Cleaner Production, 16, 2008, and J. of Industrial Ecology, 14 January 2010). To change individual consumers' behavior, 200 stakeholders proposed to increase consumers' knowledge, shift their attitudes through awareness campaigns, and change sustainability values, consumer habits, and behaviors [41].

Another EU project on Sustainable Consumption Policies Effectiveness Evaluation [42] suggested some policy measures including combination of institutional adaptations and greening of markets, new policy instruments in countries, sectors, product groups and target actors, market-based instruments which stimulate innovative, proactive companies, and underexplored market-based instruments. Radical changes are needed such as stakeholder involvement, clear multi-dimensional sustainability targets and agreements on implement-

ing steps of different agents, and implementation control with success monitoring and feedback loops [43].

The influence of power must be regarded when investigating consumption drivers, or potentials and barriers to degrowth [44]. The authors distinguish between the following elements of power: instrumental (e.g., financing lobbying or campaigns), structural (e.g., threats of corporations to shift investments and jobs to other countries), and discursive (i.e., values, norms and ideas that influence public debates and political agendas).

The project towards the Sustainable Households [45] investigated innovative household functions in three areas: clothing care, shelter (heating, cooling, and lighting) and Food (shopping, cooking, and eating). Design-orienting scenarios (DOS) included, for example, more durable and high-quality clothing, local energy providers, less supermarkets, and more small shops. Environmental, economic and consumer acceptability assessments of the scenarios have been carried out. Technological and socio-cultural changes are needed at the same time; a wide range of social and economic stakeholders in the supply and consumption chains must be active; and scenarios must be narrowly focused on consumption changes.

Many other projects have been recently carried out. Let us mention two of them. The first is the Global and Local Food Chain Assessment [46], looking at the economic, environmental, social, health and ethical performance of the food chain. The project objective was to "integrate advancement in scientific knowledge about the impact of food chains with application of knowledge to practice in order to increase food chains sustainability through public policies and private strategies". The second one, the TRANSIT [47] project, "developed a transformative social innovation theory. This is a process of change in social relations involving challenging, altering and/or replacing dominant institutions and structures. The project initiated the collaborative writing of a Manifesto for Transformative Social Innovation. With the manifesto, they wanted to redirect attention to the emerging movement of transformative social innovation: communities and individuals across the world making change on the ground".

In conclusion, SC and SP can be treated separately or together. During the production, raw-materials, energy, parts, packaging, and services are used (consumed). However, mass consumption is responsible for the production of goods. The combination of SC and SP into SCP, and integrated ways of thinking are becoming more important nowadays. SDG 12 is treating them in the same goal with 11 targets which are including waste, public procurement, reporting, information, awareness, and taxation, too.

*2.6. Education for SCP*

The spreading out of the SCP has been very much supported by educational activities. They are also important for the future awareness rising, and dissemination of scientific and practical achievements.

Petry et al. [48] have published a paper on education for sustainable production and consumption in which they apply theoretical approaches of SCP and sustainable livelihoods (SL) using five key studies. "A livelihood comprises the capabilities, assets (including both material and social resources) and activities required for a means of living. A livelihood is sustainable when it can cope with and recover from stresses and shocks, maintain or enhance its capabilities and assets, while not undermining the natural resource base" [49].

UNEP [50] has published recommendations and guidelines on Education for Sustainable Consumption (ESC). ESC methodologies are presented in the guidelines, and a roadmap of recommendations addresses the interdisciplinary issues of SCP. Governments are urged to include ESC in the education institutions, courses and curricula, teachers' training, and research. Cooperation between ESC researchers, teachers, and employers from different disciplines shall be future oriented. Creative, critical, innovative thinking, alternative lifestyles, intergenerational learning, theoretical and practical applications, social involvement, and community service must become parts of ESC.

The Regional Activity Center for Cleaner Production of Catalonia [51] has published a manual on teaching strategies for education in SCP. It aims to stimulate the creation of critical individuals who can see the relationship between socio-environmental crisis and current production and consumption patterns, and who are prepared to build a better world based on the principles of ethics, environmental sustainability, and social justice. Eight key concepts in education for SCP are described: education for SCP, direct and indirect consumptions, from cradle to grave (linear) and from cradle to cradle (circular) methods, ecological footprint, ecological rucksack, persistent organic pollutants (POPs), responsible consumption, and SCP. The SCP Workshop takes about 18 h (hours) and consists five phases: make one's acquaintance, introduction to SCP, critical analysis, putting new attitudes and behavior into practice, and evaluation of the workshop.

The UNEP Handbook [52] is explaining SCP policy. Part A presents an introduction to SCP and reviews various policy tools available to policymakers. Growing SCP trends and needs of a transition are dealt with. Experiences and proposals for SCP indicators are discussed. Part B focuses on, "policy opportunities for cleaner and safer production, resource and energy efficiency, sustainable lifestyles, sustainable cities, sustainable public procurement and sustainable tourism." Evidence for association of SDG 12 with education was found to be weak as it was not covered by MDGs [53].

The Partnership of Educators, Researchers, and Practitioners (PERL) [54] network is supported by UNESCO-UNEP. It includes more than 140 institutions in over 50 countries and is coordinated at the Hedmark University College in Norway. PERL's mission is twofold: (1) empower individuals to recognize their role as active citizens and to make more responsible daily choices, and (2) influence governments, businesses, and schools to educate individuals and to make better lifestyle choices both available and attractive.

The 10YFP on Sustainable Lifestyles and Education (SLE) program is to foster the uptake of sustainable lifestyles as the common norm with the objective of ensuring their positive contribution to addressing global challenges such as resource efficiency and biodiversity conservation, climate change mitigation and adaptation, poverty eradication and social well-being. It is based on two pillars: Rethinking how we consume and Doing 'more and better with less' [55]. Its platform on SCP is including the Hub for SDG 12 (SCP).

## 3. The Present Development and Issues of Sustainable Consumption and Production

### 3.1. Sustainable Growth or Degrowth?

Agenda 21, accepted in Rio 1992, stressed the need to change "unsustainable patterns of consumption and production, particularly in industrialized countries" (Chapter 4). At the Rio+20 conference, the unsustainable patterns have been substituted by "green growth" focusing on efficiency and innovations and so called "weak sustainable consumption". Research has shown that weak SC does not fulfill the requirements of either the Brundtland commission or Agenda 21 [42]. Practical experiences with the Paris Climate Agreement indicate that countries are hesitating to fulfill their obligations [56]. Various studies have shown that targets under the Paris Climate Agreement can only be met by developing technologies that will provide 'negative emissions' [57]—seen as a questionable strategy [58]—or that efficiency improvements, renewable resources, and low-carbon technologies with projected economic growth will overshoot such targets. It implies that slower growth, or degrowth, particularly in the developed countries, is required to stay within planetary limits [58,59]. Yet, neither the people nor the governments are ready to make radical changes in their lifestyles. This is reflected by the very slow transformation of the SCP concept, conceived at the Rio conference in 1992, into tangible action. Most policy actions thus far consisted of meetings and creating plans, rather than implementing game-changing policies [60,61]. Therefore, a "strong SC" approach is unavoidable [62].

Moving towards strong SC requires two phases of development [61]:

(1) Increase in the efficiency of consumption due to technological improvements (eco-design, sustainable production, eco-innovation, etc.), and due to more efficient use of

resources (3R—reduce-reuse-recycle, zero waste, circular economy, etc.); technically, it is a part of weak SC.

(2)    Changes in consumption patterns (habits, behaviours, and lifestyles) and reduction in consumption levels (degrowth) in developed countries requiring changes in infrastructures (what is called strong SC).

Weak SC may be e.g., using a car with a reduced gasoline usage of 3 cl/km (3 L per 100 km) while strong SC means using public transportation instead of the car. People (individuals, businesses, and governments) are ready to apply the first phase, but they are hesitant to use the second one. The main causes of overconsumption are mobility (car and air transport including holiday trips), food (meat, diary, obesity), energy use (heating, cooling, energy using appliances), and housing (building and demolition) which are causing 70–80% of the life-cycle environmental impact categories [19]. OECD countries have 19% of human population but consume 80% of the global resources. The other two groups of countries are developing ones and undeveloped ones. Their economies are different. Developing countries can leapfrog from today's situation to novel sustainable structures without copying the history of developed countries and their present consumption patterns. Underdeveloped countries have first to eradicate poverty and provide sustainable and equitable growth. Short-, medium-, and long-term impacts shall be fulfilled by businesses (production), governments (markets), consumers, citizens, and NGOs (consumption).

Sustainable development does not explicitly deny further economic growth, even in developed countries. Regarding the ecological footprint in those countries (over 6 ha per person as compared to the 1.7 ha available globally; Overshoot Day has moved from end of December in 1970 to 29 July 2019 and fell to 22 August 2020 due to the pandemic), and the increased $CO_2$ emissions (+0.6% in European Union in 2017, but 21.7% drop in emissions between 1990 and 2017) indicate that degrowth is necessary. Instead of economic growth, reduced consumption and production are needed. At the time of degrowth, the growth of wellbeing can be aimed at. It is connected to the lifestyle changes, reduced material consumption, shortened working time, and increased quality of life. The present neoliberal market system is built on constant economic growth and increased profits of the richest population. Therefore, it needs ever expanding markets and wars. The first ones require conquest of new countries, and the last ones bring destruction of existing wealth of nations and increased weapons production.

The transition from weak to strong sustainability requires a shift from the neoclassical economics to the ecological economics, from 'uneconomic growth' to improved human well-being. The precautionary principle shall be respected. Besides ecological economics, behavioral economics, sustainability marketing, environmental sociology, political science, applied philosophy, psychology, systems analysis, innovations, and historical studies are needed in the research [63].

Geels et al. [64] offer a compromise between the weak SC ('reformist' position) and strong SC ('revolutionary' position)— 'reconfiguration', based on a "transition in socio-technical systems and practices". Production and consumption are "mutually constitutive and overlapping". They suggest multilevel perspective and social practice theory approaches, "evolutionary economy, sociology technology, neo-institutional theory and social theory as theoretical inspirations", and policy mix with "gradual shift from network governance to market-based instruments and regulations". Additionally, they are presenting critical thoughts about future SCP research programs.

Degrowth became a "political, economic, and social movement based on ecological economics, anti-consumerist and anti-capitalist ideas" [65]. It is also considered an "essential economic strategy responding to the limits-to-growth dilemma" [66]. Nine international degrowth conferences have been organized, so far—the initial one in Paris (2008 with 140 participants), the 1st one in Barcelona (2012) and the 8th one in The Hague (2021, the 6th one with 600 participants). Research and Degrowth [67] is an "academic association dedicated to research, training, awareness raising and events organization around degrowth". Further information can be found in Demaria [68]. Essential trends for degrowth [69] are:

- "Striving for the good life for all; it includes deceleration, time welfare, and conviviality.
- A reduction of production and consumption in the global North and liberation from the one-sided Western paradigm of development. This could allow for a self-determined path of social organization in the global South.
- An extension of democratic decision-making to allow for real political participation.
- Social changes and an orientation towards sufficiency instead of purely technological changes and improvements in efficiency to solve ecological problems. They believe that it has historically been proven that decoupling economic growth from resource use is not possible.
- The creation of opened, connected, and localized economies" (i.e., strong SC).

*3.2. Measuring Sustainable Consumption and Production*

Economic development of countries or regions is usually measured by Gross Domestic Product (GDP) increase [70]. To level off disparities in the cost of living and inflation rates, GDP per capita at purchasing power parity (PPP) is used to compare living standards between nations. However, GDP is a poor measure of social progress, and it does not take harm done to the nature into account.

"Economic indicators such as GDP were never designed to be comprehensive measures of prosperity and well-being" [71]. "The EU's Beyond GDP initiative is about developing indicators that are as clear and appealing as GDP, but more inclusive of environmental and social aspects of progress". In 2007, the European Commission, European Parliament, Club of Rome, OECD, and WWF (World Wildlife Fund) hosted the high-level conference Beyond GDP. "The objectives were to clarify which indices are most appropriate to measure progress, and how they can best be integrated into the decision-making process and taken up by public debate". In August 2009, the European Commission released its road map "GDP and beyond: Measuring progress in a changing world". In August 2013, the European Commission published the paper "Progress on GDP and beyond actions" [72]. Some indicators have been developed but no breakthrough has been achieved, so far.

Human Development Index (HDI) [73] has been constructed to evaluate the social development component. The HDI is a "composite index of life expectancy at birth, adult literacy rate and standard of living, adjusted to PPP". The most developed countries in Europe and North America are on the top, again. The World Happiness Report [74] "combines GDP per capita, social support, healthy life expectancy, freedom to make life choices, generosity, and perceptions of corruption". Developed countries are leading the ranking. The same is true for the OECD Better Life Index: OECD has elaborated Green Growth Indicators, but they are not included in any sustainability index [75].

The Happy Planet Index (HPI) has been the only one to properly include "human well-being and environmental impact" [76]. "The usual ultimate aim of most people is not to be rich, but to be happy and healthy". However, "sustainable development requires a measure of the environmental costs of pursuing those goals." Therefore, higher scores are ascribed to nations with lower ecological footprints. The HPI combines the following four elements: wellbeing, life expectancy, inequality of outcomes, and ecological footprint (EF). The first three elements are in the numerator and the last one is in the denominator.

The developed countries are doing well in all the three elements of the numerator, but they are bad in the denominator (the ecological footprint). The EF measures biologically productive area available within a region or the world (bio-capacity). The world average per capita is 1.7 ha but global consumption reached 2.8 ha in 2014. "Today, more than 80% of the world's population live in countries that are running ecological deficits, using more resources than their ecosystems can renew" [77]. "Humanity uses the equivalent of 1.7 Earths to provide the resources we use and absorb our waste. We use more ecological resources and services through overfishing, overharvesting forests than nature can regenerate, and emitting more carbon dioxide equivalent into the atmosphere than forests can sequester."

Developed countries, "do not rank highly on the Happy Planet Index. Instead, several countries in Latin America and the Asia Pacific region lead the way by achieving high life expectancy and wellbeing with much smaller ecological footprints". Costa Rica is the best country. Norway is 12th, Albania 13th and Spain 15th. UK is at the 34th place, France at the 44th, and Germany at the 49th one. United States rank is 108th, Russia 116th, and Luxemburg 139th. A plot of HPI versus GDP per capita revealed that the highest HPI value (about 55, in South-East Asia and China) was achieved at the GDP per capita (PPP) equal to 4500 USD; the curve is very steep at lower values (being the lowest at $I_{HP} = 25$, with Central and Southern Africa, and new EU member states), and it is falling slowly towards the richest region, North America ($I_{HP} = 30$). "Happiness does not need to cost the Earth. People can live long, happy lives without using more than their fair share of the Earth's resources/sinks per capita."

The global SCP indicator framework was developed [78], adopted by the General Assembly in 2017, and approved yearly by the UN Statistical Commission in 2018–2021. Indicators of the Goal 12 with 8 targets and 3 target actions are listed in reference [79]. The progress by each target indicator can be followed at the UN SDG 12 Hub [80].

Eurostat is reporting on SDG 12, Responsible Consumption and Production using seven indicators, grouped into three areas: decoupling environmental impacts from economic growth (the European Green Deal), the green economy (environmental goods and services sector), and waste generation and management [3]. Key trends in the last five years regarding the EU targets are: significant progress in energy productivity and green economy, moderate progress in resource productivity and circular material use rate, insufficient progress in consumption of toxic chemicals, $CO_2$ emissions from new cars, and generation of waste (excluding major mineral wastes).

Several other sets of indicators exist. Let us mention the early ones [81] and the most recent one [82]. The SDG Tracker [83] measures progress towards the SDGs all over the globe. Eurostat [84] is monitoring progress towards all the SDGs in an EU context.

### 3.3. Existing Trends in Sustainable Production

The industry tried to cope with the present problems of SCP by improving its efficiency. Some examples will be shown in this section such as resource efficiency, circular economy, eco-design, the Green Industry approach, and European chemical industry efforts to decouple economic growth from resource use.

Resource efficiency means using the Earth's limited resources in a sustainable manner while minimizing impacts on the environment. It allows us to create more with less and to deliver greater value with less input [85].

Circular economy, "aims to redefine growth, focusing on positive society-wide benefits [86]. It entails gradually decoupling economic activity from the consumption of finite resources and designing waste out of the system. Underpinned by a transition to renewable energy sources, the circular model builds economic, natural, and social capital. It is based on three principles: (1) Design out waste and pollution; (2) keep products and materials in use; and (3) regenerate natural systems. Transitioning to a circular economy does not only require adjustments aimed at reducing the negative impacts of the linear economy. Rather, it represents a systemic shift that builds long-term resilience, generates business and economic opportunities, and provides environmental and societal benefits". "In a circular system, resource input and waste, emission, and energy leakage are minimized by slowing, closing, and narrowing energy and material loops. This can be achieved through long-lasting design, maintenance, repair, reuse, remanufacturing, refurbishing, recycling, and upcycling" [87].

Eco-design enables production of goods and service by meeting the needs of customers while using minimum resources possible, having a minimum impact on the environment and society. The designer must maximize the use of renewable materials, minimize the energy use, and enable the product to be reused or recycled at the end of its lifecycle. He must follow the existing standards and legal requirements. The EU's directive on

eco-design of energy-using products [88] "provides a framework for establishing minimum requirements for every-day products that account for a large proportion of total energy use. Measures include ambitious product standards for eco-design, green public purchasing, eco-labelling, and eco-innovation. The European Commission plans to extend compulsory eco-design requirements—already in place for 'energy-using' products such as household appliances—to 'energy-related' products that have environmental impacts during their use (for example, windows or water-distribution devices)." [89].

In the last decade, the concept of Green Industry was developed to match sustainable development in industry with the global challenges of sustainable development [90]. The initiative is raising awareness, developing knowledge, and building capacity. They cooperate with governments to support industrial institutions, providing in turn assistance to enterprises and entrepreneurs in all aspects related to the greening of industry. The initiative is working within the following sectors:

- Resources (raw materials, energy, and water) efficiency
- Clean energy and cleaner production
- Low emissions and climate friendly solutions
- Responsible agriculture
- Chemical leasing
- Corporate social responsibility
- Stakeholders' engagement

An example of future thinking is the European Technology Platform (ETP) Sustainable Chemistry [91], a forum that brings together industry, academia, policy makers, and the wider society. It was established in 2004. Chemical industry contributes about 20% to the European gross value added.

SusChem's Strategic Innovation and Research Agenda recognizes "three overarching and interconnected challenge areas: circular economy and resource efficiency, low carbon economy towards mitigating climate change, as well as protecting environmental and human health. They will require sustainability assessment innovation, education and skills capacity, and enabling digital technologies". The SusChem technology priorities are summarized under the following main chapters: advanced materials, advanced processes, enabling digital technologies, and horizontal topics. "Responsible consumption and production will be achieved through improved process efficiency, utilization of alternative feedstock, sustainable management and efficient use of natural resources, including critical raw materials circularity and the valorization of waste."

European chemical industry has been successful in fulfilling the planned actions and applied research results in practice [92]. In the period from 1991 to 2018 it achieved good results:

- GHG emissions were reduced (−49.6%) and decoupled from the increasing chemicals production (+94.7%)
- Specific GHG emissions were reduced by 42% per energy consumption and fell by 74% per production.
- Fuel and energy consumption was reduced by 24% since 1991, and energy intensity (GHG emissions per production) fell by 55.7%.
- Renewable energies consumption has doubled since 2000.

## 4. Sustainable Consumption and Production in Future

The future is impossible to predict. However, scenarios can be forecast together with their impacts and activities needed. Some trends can be used to forecast SCT in the near future. We need to know the most important global trends (megatrends) to understand the future development, adapt and control the SCT development. Research and development, innovations and spirit of enterprise are vital for preparing and executing the changes needed. We also need to know how the human development is evolving regarding the technological innovations; using the wavelike behavior of human development enables us to plan the transition activities and adopt the coming changes. For example, we know that

GHGs rise, and population growth increase the pressure on resources, natural environment, and social relations. Therefore, we must plan activities to reduce the GHGs as well as population growth and consumption per capita at the same time.

### 4.1. Megatrends

"Megatrends are large, social, economic, political, environmental or technological changes that are slow to form but which, once they have taken root, exercise a profound and lasting influence on many if not most human activities, processes and perceptions [93]." "They are large, transformative global forces that impact everyone on the planet" [94]. The most known megatrends were published by PricewaterhouseCoopers [95], Ernst and Young [94], the European Commission [96], the National Intelligence Council [97], and the European Environment Agency [98]. "The global EEA megatrends report assesses 11 global megatrends (GMT) of importance for Europe's environment in the long term [2]." "In assessing key drivers, trends and implications for Europe, it aims to provide an improved basis for strategic European environmental policymaking." WEF elaborated "six technology mega-trends shaping the future of society" [99].

The twelve most cited megatrends (four societal, *four economic* and four environmental ones; the most often cited ones are printed in bold) and possible solutions are summarized here:

1. **Demographic and social changes** (population growth, ageing; poverty, inequality, migrations) with diverging global population trends (fertility, mortality)—education, and solidarity.
2. Health management and cost pressures, changing disease burden, and risks of pandemics—healthy living, public health systems, and intensive research.
3. **Rapid urbanization** (megacities, mobility, security)—smart: cities, communities, and homes.
4. Regional instability (public debts, crises, economic and financial shocks, migrations, conflicts and wars, danger of collapse)—increasingly multipolar and smart world.
5. *Shift in global economic power from G7 to E7 (China, India, Brazil, Mexico, Russia, Indonesia, Turkey), shift to networks and coalitions in a multipolar world—empowerment of individuals.*
6. *Limits to continued economic growth, global marketplace, entrepreneurship rising—beyond GDP initiative, human capital, happiness, and degrowth.*
7. ***Resource scarcity*** *(water, critical raw-materials, fertile land, forests) and intensified global competition for resources (prices volatility, potential conflicts)—dematerialization, resource efficiency and circular economy.*
8. *Accelerated **technological breakthroughs** (Industry 4.0 with digitalization, artificial intelligence (AI), Information and communication technologies (ICT), robotics, nano-, bio-, and eco-technologies, renewables, health care, low-carbon solutions, microbiota and synthetic biology—education, innovations, Industry 5.0).*
9. Growing pressures on ecosystems (population, food and energy consumption, water scarcity, mobility, decline in biodiversity), and hysteresis effect—decoupling growth from resource usage.
10. Increasingly severe consequences of **climate change/crisis** (global warming, deforestation, desertification, natural disasters, extreme weather events)—adaptation, mitigation, carbon capture and storage, and degrowth.
11. Increasing environmental pollution load (air pollution, land releases of nutrients from agriculture and wastewater, and water and marine pollution)—sustainable consumption.
12. Diversifying approaches to governance (due to globalization, governments are facing a mismatch between long-term, global, systemic challenges facing society, and their more national and short-term focus and powers)—social cohesion, and policy makers' cooperation.

Societal uptake of new technology is speeding up—while 46 years were needed for the prevalence of electricity amongst 25% of USA population, it was 31 years for radio, 13 years for mobile phone, and only 7 years for the World Wide Web.

The Paris Agreement on GHG emissions reduction to keep the temperature rise below the 1.5–2.0 °C above the pre-industrial levels was signed by 195 countries. The EU is in the forefront; its climate change action contains the following key targets:

- GHGs emissions reduction as compared to 1990: 20% by 2020, 55% by 2030, and net zero by 2050.
- Increase fraction of renewable energy consumption to 20% by 2020, and 38–40% by 2030.
- Rise energy efficiency: 20% by 2020, and 32.5% by 2030.
- Use less water by adapting building regulations, flood prevention, and developing crops that cope better in drought conditions.

Financial support and regulation are planned to be used to achieve the targets. Climate research, education, awareness rising, NGOs activities and extreme weather events can speed up the transition, too. Similar agreements are needed for population stabilization, species extinction prevention, land and other resources use, waste reduction (to zero), dangerous waste, and pollutants.

Many associations, societies, and platforms have elaborated their visions and trends. The US National Academy of Engineering (NAE) together with the Chinese Academy of Engineering (CAE), and the UK Royal Academy of Engineering (RAE) organized three Global Grand Challenges Summits (GGCS), the third one in July 2017. They have proposed the 14 'game-changing goals' [100]:

1. Advance personalized learning
2. Make solar energy economical
3. Enhance virtual reality
4. Make computers to process information like humans do
5. Engineer better medicines
6. Advance health informatics
7. Restore and improve urban infrastructure,
8. Secure cyberspace
9. Provide energy from fusion
10. Prevent nuclear terror
11. Supply clean water to all
12. Manage the nitrogen cycle
13. Develop carbon sequestration methods
14. Engineer the tools of scientific discovery

*4.2. Wavelike Human Development*

Human development is going on in cycles of different durations and characteristics. We are now leaving the industrial era by entering a postindustrial one. The industrial era has been characterized by several waves or cycles. There are two most accepted classifications of them:

- Four industrial revolutions [101] with the top prosperity year shown:
    1. Mechanization, steam, and water-power usage (mechanical automation), 1765
    2. Mass production, assembly line, electricity (industrialization), 1870
    3. Computer and automation (electronic automation), 1969
    4. Cyber-physical systems (Industry 4.0 or Factory 4.0, smart automation,), now

- Six Kondratiev's innovation cycles, lasting 45–60 a (years) each [102]:
    1. Cotton, iron and hydropower, duration, 1780–1848
    2. Railways, steam power and mechanization, 1848–1895
    3. Steel, heavy engineering and electrification, 1895–1940
    4. Oil, cars, and mass production, 1940–1980

5. Information and communication technology, 1980–2015
6. Clean technology and resource efficiency? 2015–2045?

Each wave has four phases: growth, prosperity, recession, and depression.

In Industry 4.0 (also called smart factory, factory of the future, or Cyber-Physical factory), "computers and automation will come together in an entirely new way, with robotics connected remotely to computer systems equipped with machine learning algorithms that can learn and control the robotics with very little input from human operators [103]." For a factory or system to be considered Industry 4.0 (a responsive, adaptive, and connected manufacturing) it must include: "interoperability, information transparency, technical assistance, and decentralized decision-making". Industry 5.0 is supposed to be about cooperation between people, and smart systems and machines (collaborative robots using artificial intelligence). Employees will provide value-added, creative tasks to enable mass customization and personalization for customers, e.g., in production, medical care, life-long product surveillance.

"The 6th wave of innovation will be about resources—natural resources, human resources and information; it is heralded by massive changes in the market, societal institutions and technology that all reinforce each other [102]." The authors believe that humans will diminish resource dependence by increasing resource efficiency but degrowth may be needed. Waste is an opportunity (for circular economy), and nature is a source of inspiration; service shall be sold, and not the product; digital and natural will converge. There will be a "spectacular boom in technologies ranging from clean technology to digital mapping to online collaboration. We are moving from an old mode of operation when we were harvesting resources that were plentiful and cheap to a time when we are managing resources that are scarce and valuable."

The 7th wave may be about human efficiency, human capacity for working longer, healthier, better, and smarter.

### 4.3. Important Future Trends and Adaptations in SCP

In the future, demands for SCP will follow the life cycle of products and services. The life cycle starts with raw materials extraction and energy to enable production, continues with consumption, and ends with waste prevention and recycling [104]. Using natural resources more efficiently enables healthier lives, saves money, creates jobs, and respects the limits of the planet. The EU intends to be "resource-efficient, green and competitive low-carbon economy"; this is one of the three objectives of its 7th Environment Action Program [105]. The Roadmap to Resource-Efficient Europe aims to use "the Earth's limited resources in a sustainable manner while minimizing impacts on the environment". But resource efficiency will not be enough, and we must reduce resource use growth substantially.

"In the past 50 a (years), 60% of the Earth's ecosystem has been depleted, and natural resource consumption is expected to rise 3–6 times by 2050" [106]. "The population is expected to reach over 9 billion people by 2050, and the global middle class is expected to triple by 2030. With this in mind, we must ask: how long can we sustain this development model without further damaging the environment and aggravating existing inequalities?" The main predictor of the level of environmental pollution through consumption is related to the income per capita.

#### 4.3.1. Sustainable Consumption at a Turning Point

Evidently, the transition from weak to strong sustainability and degrowth will be needed. The possible solutions must be studied and enlarged further to enable the transition in the next 30 years. SC will be achieved by two major paths: the pressure of legislation, and education for better understanding and adapting to the changes. Consumption taxes on goods and services that embrace sales tax, value added tax, excise tax, and expenditure tax will be increased and additional emission tax introduced. It is also important to reduce and remove subsidies on goods and services that are causing unsustainable behavior— environmental, social, or economic ones. Green public procurement is one of the methods

which can include SCP friendly conditions. Standards, e.g., on how long products should last or how easy they should be to repair and recycle, also support SC.

Extremely important is education for sustainable development (ESD). Besides the formal and lifelong education, transparent information to consumers enables them to make more sustainable choices. The European Commission has developed policies and tools to help identify green products and reward sustainable production practices, e.g., ecolabels, energy efficiency labels, product environmental footprints, information on packaging and packaging waste, and on "resource efficiency in the building sector" [107]. The Retail Forum is one of the ways to promote more environmentally friendly and energy-efficient products as well as provide better information to consumers on how to use products in the environmentally and economically most efficient way. Education for degrowth is also needed.

In the first phase, weak sustainability, it is of utmost importance to maintain the social market economy model, not the neoliberal approach to prevent social revolts of poor people, revolutions, and wars. Distributed and sharing economy are being introduced to reduce individual consumption and pollution. Circular economy enables products, materials, and energy to be reused, remanufactured, refurbished several times before recycling them to raw-materials or energy users. Smart cities and communities concept includes: "applied innovation, better planning, a more participatory approach, higher energy efficiency, better transport solutions, intelligent use of ICT, etc." [37].

Lifecycle thinking, design for environment, and design for efficiency shall provide more efficient equipment and systems, and include integration and optimization of networks, supply chains, value chains, and logistics. Life cycle approaches and tools have been developed to be used in the private and public sector. Life Cycle Initiative (LCI), hosted by UN Environment has elaborated the Life Cycle Approach (LCA) to assist in decision-making at all levels of product development, production, procurement, and final disposal. They have accepted a strategy for the years 2017–2022 [108].

"Gaps between the forecasting decisions of producers and the consumer demand create an estimated 30% waste in manufacturing goods" [106]. "Internet of things will enable companies to track products for supply-chain purposes and decrease waste. Companies are starting to incorporate data and analyses provided by this new industrial revolution. They are producing better products with maximum societal value while minimizing environmental cost. Consumers will be able to choose and use their products more efficiently and sustainably. They will also need to reevaluate their lifestyle and their environmental, social, and economic impacts. They will need to assess how they choose and use products and services, and change their own consumption patterns. Millennial consumers (eco-natives) are increasingly looking for products that make them look and feel good, and which are also good for the planet and society. Governments and civil society will also need to engage and encourage the removal of unsustainable products and services from the marketplace." WEF [106] is reporting about some encouraging achievements in this area.

The second phase, strong sustainability with resource usage degrowth is still to be agreed upon, and we are lacking the time to do it before it will be too late. "The world's leading climate scientists have warned there is only a dozen years for global warming to be kept to a maximum of 1.5 °C, beyond which even half a degree will significantly worsen the risks of drought, floods, extreme heat and poverty for hundreds of millions of people." [109]; until 2030 emissions had to be reduced by 55% below the 1990 level. Education, awareness raising, youth engagement, and scientific research, as well as natural disasters can speed-up the transition. To address the over-consumption, it is not enough to rely on technological measures. Strong economic measures will be necessary, and maybe even a transformation of the economic system will be required.

Let us take EU as an example. The 27 member states have reduced their GHG emissions from 4.9 Gt in 1990 to 3.7 Gt/a ($10^9$ or billion tons per year) in 2019; they have reached the 2020 target of −20% reduction two years in advance. If they want to achieve the 2030 target of 55% reduction, they must lower the GHG emissions to 2.2 Gt/a [98], and

to net zero by 2050. The population in EU is increasing very slowly (0.16%/a), therefore, the reduction must be achieved by increased efficiency and reduced consumption. It requires degrowth.

Degrowth of GHG emissions can be achieved by substituting nonrenewable energy sources with renewable ones (solar, wind, biomass, geothermal) and increased efficiency. Degrowth of material usage can be accomplished by circular economy ('growth within') using the 12 Re-s (Rethink, Reform, Refuse, Reduce, Reuse, Re-Gift, Repair, Refurbish, Remanufacture, Repurpose, Recycle, and Recover) while achieving zero waste. These two activities can be accompanied by increased happiness which is including reduced stress, improved equity, social relations, culture, health, and well-being. But these modifications require radical changes in habits and lifestyles.

4.3.2. Sustainable Production Trends

Sustainable production shall depend on the sustainable consumption, not vice versa as it is the case now. Degrowth is expected in near future. It shall be based on Lifecycle Thinking which is "made operational through Life Cycle Management (LCM)". LCM is including [110]:

- Systems and procedures, such as communication, eco-labelling, sustainable procurement, environmental management systems, dematerialization, environmental impact assessment.
- Data, information, and models, e.g., databases, best practices.
- Tools and techniques, such as Life Cycle Assessment (LCA), Life Cycle Costing (LCC), Cost-Benefit Analysis (CBA), Input-Output Analysis (IOA), Cleaner Production Assessment (CPA), increased resource (energy, water, materials, employees, and finance) efficiency, and circular economy will be the most important goals for future sustainable processing and manufacturing.

Waste minimization towards zero is another important target in SCP. Smart mobility with second generation biofuels, electrochemical cells and batteries, novel combustion and gasification technologies, advanced energy systems (renewable sources, combined heat and power, heat pumps, and poly-generation), and carbon capture, storage and reuse are needed to minimize and replace fossil fuels.

New raw material base for process industries (biotechnology, using biomass and waste, photosynthesis from $CO_2$ using algae or inorganic synthesis) will be very important. Biotechnology is developing new processes, new raw materials, and more sustainable application of resources. Synthetic biology, microorganisms and biomimetic catalysts are used for raw materials and solar energy production. The market potential of bionics in medicine is increasing rapidly. Bioengineering and smart automation will be leading the development.

Sustainable and intelligent process design, life-cycle planning, eco-design, process modelling and simulation, computer aided design (CAD) and control (CAC), optimization of processes using mathematical programming, such as mixed integer (non)linear one (MINLP), multi-objective optimization, process intensification, product analysis, synthesis and design, product/service optimization are some of the promising tools for sustainable production. Energy, water, mass and waste integration of processes, and industrial symbiosis (eco-industrial parks) are to increase resource efficiency, reduce costs, emissions, and pollution. Product and process safety, risk reduction, public and occupational health, improved regulations and legislation shall be protecting employees and product users.

WEF Report on six technology mega-trends [111] argues that the technology-enabled shifts will "provide two promises: (a) digital connectivity for everyone to everything, anywhere and at any time; and (b) tools for analyzing and using digital data in new ways." The report groups the twenty-one shifts discussed into six mega-trend categories: "(1) people and the internet; (2) computing, communications and storage everywhere; (3) The internet of things; (4) artificial intelligence and big data; (5) the sharing economy and distributed trust; and (6) the digitization of matter."

Important sources of information about the future development are international research projects, e.g., Horizon ones [112], science and engineering conferences (e.g., [113]), international forums and shows (e.g., ACHEMA, Ausstellungstagung für chemisches Apparatewesen), and European technology platforms covering bioeconomy, energy, environment, ICT, production and processes, and transport [114]. The 2018 ACHEMA participants discussed about value chains, photo-redox catalysis, zero-waste strategy, and digitization; its congress concentrated on flexible production, industrial and laboratory safety, materials technology and testing, pharmaceutical technology, process analytics, raw materials, and water technology. The 2018 ETP SusChem stakeholders focused on advanced materials (circularity, energy storage, production and efficiency, and functionality and performance), advanced processes (digital technologies for process design and control, bio-based and $CO_2$ feedstocks), as well as sustainability assessment, skills, consumer awareness, and education and training.

Digitalization will create enormous opportunities in companies and will, "increase customer value by streamlining processes, increasing quality, creating new revenue streams, and reducing production costs". Basic concept of digitalized (smart) industry will include [115]: big data, internet of things and services, artificial intelligence, logistics, business model innovation, and horizontal and vertical integration. Advanced concepts will deal with quality management, maintenance, aftersales service, business analytics and data mining, robotics, and autonomous systems. "The digital factory will be based on human-machine collaboration, augmented reality, cloud computing and assistant systems, change management, IT security, continuous and digital engineering, active manufacturing and 3D printing, and digital implementation management." "Cyber physical systems (CPSs) have the potential of substantially changing the way we are currently constructing technical systems, and the way we are interacting with our environment in many domains; the five functional areas that have validity across enterprises and different sectors of industry are: data collection and processing, assistance systems, networking and integration, decentralization and service orientation, and self-organization and autonomy." [116].

The potential and the challenges are huge, but at the same time there are real concerns (for example, regarding privacy).

## 5. Conclusions

Different visions exist on the solution of unsustainable production and consumption. There is increasing world population and growing consumption per capita on one side, and depletion of natural resources, pollution, climate change and species extinction on the other one, both requiring serious changes in human behavior. Sustainable Development Goals (SDGs) are a very important guide for sustainable development in the future. According to SDG 12, SCP can help in building environmentally sound, socially acceptable, and economically viable development. It has made an interesting evolution since the UN conference in Rio de Janeiro until it has been established as one of the 17 SDGs. Increased regulation and "better education for sustainable development" are the most important tools. Sharing economy, smart cities and communities, lifecycle thinking, and artificial intelligence can improve unsustainable consumption. Circular economy, increased resource efficiency, waste reduction, renewable sources of energy and raw materials, and carbon capture, storage, and reuse will be the most important activities in sustainable production.

There is no doubt that digitalization, artificial intelligence, computers, automation, and robotics will have an enormous influence on consumption and production. Computer aided planning, design, operation, and optimization will enable humans to produce more with less and recycle materials and energy while better respecting the needs of consumers. Industry 4.0 or smart factory along the whole value chain is a continuation of the ICT but it will need an upgrade with net-zero GHG emissions, zero waste, and zero pollution. The present concept is undervaluing the environmental and the social pillars. Degradation of the environment (population and consumption per capita growths, climate change, resource depletion, urbanization, desertification, deforestation, etc.) are proceeding much

faster than planned. The 6th wave theory is much broader than the Industry 4.0 one. It aims to "decouple economic growth from resource consumption" and increase resource efficiency, fueled by institutional changes such as carbon pricing and accelerated by clean technologies [102]. Degrowth of resource use will be obligatory in the future. Therefore, we must stimulate research and development, innovations, and entrepreneurship. The rate of technological development is exponential and proportional to the number of brains on the planet; the more people we have, the faster we innovate [117].

Although the pace of change is slowing down, the absolute growth is still going on and the hysteresis effect is endangering humans' future. We must do much more than currently planned with the Paris Agreement, and the EU action plans. Lowering of GDP per capita towards the 4500 €, and respecting happiness with profound influence on the habits and the way of living in developed countries are needed. According to the Europe 2020 strategy, we must ensure food, energy, water and raw materials security, clean air and waters, resource, and transport efficiency, and climate action. Wars, tax havens, food waste and obesity shall be forbidden by law, consumption shall be taxed much higher, and lifelong learning shall be intensified.

Social non-equalities are growing with intensification of the neo-liberal economy model since the mid-seventies of the last century. A world where 1% of humanity controls as much wealth as the other 99% will never be stable [118].

Profits and succession duties shall be taxed much higher, social security, health insurance and pensions made accessible and affordable to all, equalities enacted, unemployment lowered, and poverty and hunger abandoned. The best way to predict the future is to create it (Peter Drucker, in [102]). The Sustainable Development Goals and targets together with Sustainable Consumption and Production No. 12 shall be realized by 2030.

**Funding:** This research received no external funding.

**Institutional Review Board Statement:** Not applicable.

**Informed Consent Statement:** Not applicable.

**Data Availability Statement:** Not applicable.

**Conflicts of Interest:** The author declares no conflict of interest.

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
