# Peer review of "Evolution and Current Challenges of Sustainable Consumption and Production"

_sustainability, doi:10.3390/su13169379_

Round 1

Reviewer 1 Report

The article is a very in-depth research, particularly for a journal article and the reader interested in the subject can learn a lot from it. The author cited a clear review of literature, appropriate references were used in the introduction section. The statements contributed to the overall understanding of the subject. Section titles are appropriate. The purpose was clearly and concisely stated and agreed with the title. 

It was well written and well organized. Overall, it was a very interesting, significant contribution to the field of research. 

Author Response

Thank you very much for the extremely positive review.

Reviewer 2 Report

This manuscript addresses an interesting policy trend in SCP policy. The topic truly fits the special issue and would enhance the various views in the special issue. However, the structure of this manuscript is strange for a scientific article and seems not appropriate for the author’s intent. Focuses of discussion points are not so clear, and time to time it deviates from the main point of each section/subsection. The author mixes the results of the review and the author’s claims, which deteriorate the quality of the valuable review of this article to some extent. Specific comments are as follows.

  1. Abstract:

A bit lengthy and the points are not so clear. The main purpose of the abstract is to draw attention of readers so that they become inclined to read the main text. I highly recommend elaborating the current abstract.

  1. Lengthy or duplicated explanation

Overall, explanation is redundant and duplicated beyond a single subsection, which could bring the loss of readers interests.

Make the manuscript as concise as possible overall.

  1. Line 39, “we are not satisfied with the results achieved”

“We” is not appropriate to use in scientific articles except for the case that it means the authors. Specify the subject of this sentence or rephrase this sentence. In addition, “satisfy” is very subjective and unclear. Reconsider this expression so that this sentence explain the fact in a clearer and objective way.

  1. Lack of the objective in Introduction

Any scientific articles including review articles MUST include the clear explanation of the objective of the article in Introduction. What is Section 2 of the current version of the manuscript is unclear. A kind of introduction but separated from Section 1. A kind of a review result but also seems just an explanation of key concepts of this article.

Please clearly explain it the latter part of the introduction such as

“The objectives of this article is therefore ....”,

“I therefore reviewed ... to ...”, or whatever.

Subsection 2.8 is placed strangely. It should be moved to the Introduction section so as to fit the standard writing style of scientific articles. The aim of scientific articles should be explained in the Introduction.

The author may take the explanation in Section 2 for granted and believe that Section 3 is the main part of this article; however, it is not for the readers and the author has to explain what is Section 2 and what is Section 2 distinguishing them. I strongly recommend to explain the structure of this article in Introduction (before Section 2) such as “The object of this article is .... This article starts with the overview of ... in Section 2 and then move onto results of reviewing ....”

Usually, the method is explained in an independent section (usually Section 2). For this article it can be explained just after the abovementioned objectives and structure in Introduction.

  1. Section 2.1

This section has four drawbacks.

First, in this section, as the author explains the history, the sentences should specify the year of the event. Especially, at Line 120, the year of the Johannesburg Plan should be explained for readers who are unfamiliar with it.

Second, explanation about SCP around Rio+20 is insufficient or unclear for readers. It seems for me that 10YFP is suddenly explained though I understand this section focus more on the explanation of the concept rather then SCP history.

Third, the paragraph between Line 128 and 133 explains about the academic community. However, the references referred to are limited to those published about after 2005. In this sense, the author fails to explain the overall responses from the academic community. Add more references in the 1990s or reconsider the beginning of this paragraph so that readers can understand the authors explain the recent discussion in the academic community.

Forth, the heading of this section does not fit to its main text. The birth of SCP connotates things happened in the 1990s, but the current explanation extends the time horizon to the present. Reconsider the structure of Section 2.

  1. Position of Section 2.3

Section 2.3 can be merged into Section 2.1 as the explanation about the birth necessitate the explanation of the definition. In other words, Sections 2.2, 2.4, 2.5 explaining about historical events are split strangely.

  1. The order and necessity of subsections in Section 2

I do not understand what is the rationale of the order of the subsections in Section 2. Moreover, the current manuscript lacks explanation about in what order several concepts and aspects will be explained in Section 2.

I recommend to reconsider the order of subsections in Section 2 because it is a bit difficult to understand.

In addition, in Line 405, the author pays attention to factors for transformation. It is understandable and to the point for the current challenges of SCP; however, does Section 2 properly explains the importance of transformation and guide the readers to the author’s intention of this articles and to the Sections 3 and 4?

Not only reconsider the order but also reconsider the necessity of each subsections in Section 2. It does not effectively connect to Sections 3 and 4 in the current version of the manuscript.

  1. Subsection 2.8

As commented earlier, please reconsider the location of Section 2.8 in this article. In addition, the subsection is a bit lengthy and should be shortened. Mere listing the documents does not justify the coverage of the reviewed articles much. So, the author can delete the list and just mention “reviewed reports published from international organizations, ..., ..., ..., and so on.” simply.

The last paragraph in the current Section 2.8 should be mentioned in Section 4. It is too detailed for what is explained before Section 3 starts.  

  1. Lines 400-401, “The paper has been considerably improved on the bases of reviewers’ comments and suggestions”

Inappropriate. Delete it. The author does not have to and should not mention review process as all the article published in the journal is reviewed by anonymous reviewers.

  1. Section 3.1

Circular economy and resource efficiency suddenly appears in this section; however, it is irrelevant in the context of the current version of Section 3.1. Delete it. (or explain why CE and RE are important topics for this article and how it is relating to the conclusion of this article.)

Circular economy and resource efficiency are definitely relevant key concepts for SCP. So the authors may argue them elsewhere appropriate.

Remove the bold in the text. It is not appropriate for scientific articles. It seems borrowed from a report or the like.

  1. Section 3.2

Indicators are an important tool; however, why do the author need to review it in light of the objectives of this article? Without such explanation, this section should be removed. I do not think that the author concludes anything valuable from the review of the indicators.

Table 1 should be deleted. Few discussion is made in this article and no valuable information can be derived from this table. The SDG indicators are already discussed by others in scientific articles. You do not have to explain the superficial information in this article.

  1. Section 4

This section does not seem scientific compared to the other sections because the author mixes the results of the review and the author’s claims and opinions. Be cautious to distinguish them (preferably, the review results and the author’s claims should be written in different subsections.) and reduce the unnecessary claims of the author’s.

  1. Necessity of subsections in Section 4

As I commented for Section 2, reconsider the necessity of each subsections in Section 4, too. I do not understand the rationale of the structure of this section. The current explanation is fragmented and hard for readers to follow.

  1. Subsection 4.2

Shouldn’t this subsection be discussed in a sub-subsection of Section 4.3? It seems an important aspect of future SCP, technology and industry.

In addition, I do not clearly understand the point of discussion in this subsection. The first half explains the cycles and the second half explains the phase of industry. In addition, these are presented independently. It is not necessary to show off what the author knows. Focus on a discussion point in each section/subsection and articulate the series of explanation. Explain how these explanations lead to your conclusion. Delete unnecessary explanations that do not linked to the conclusions are the research objectives. I recommend to delete this subsection because it does not provide an original aspect of this article about the cycles and industry phases. If the author does not to delete, explain what is a new contribution of this part?

(In Lines 623-625, the author asserts “we need to know ...” but I do not understand why we need to know and what is the benefit of knowing. Lacks of explanation.)

  1. Subsection 4.3

This must be the discussion part of the author. Reconsider the heading of this section to convey this.

What is the answer to the question “When can we expect the turning point and what are the decision factors for this transformation?” which are included in the aim of this study?

Questions and aims presented in advance are not discussed in this section. Better linkages with the aims and conclusions are necessary.

In addition, in Line 761, “Sustainable consumption will follow two major paths: legislation and education ones”. But are regulatory and educational approach for strong sustainability and transformation? These two paths seem conventional SCP approach and no longer sufficient for SCP at present and in the future. If “will follow” is replaced by “has been attempted to achieve two major paths”, it is understandable.

Why does the author argue SC (consumption) and SP (production) independently? Doesn’t the combination and integrated ways of thinking become more important nowadays?

Overall, the impression of the arguments in this section is lengthy and not focused. Make it concise. Showing the discussion points, “first, ..., second, ...” is a good way to guide readers in your discussion.

  1. Conclusions

Conclusion is not based on the results and discussion in the previous sections. The authors still keep arguing from the previous section. Wrap up the main points and emphasize the key messages. Entire rewriting is necessary.

Round 2

Reviewer 2 Report

The heading of the last section was weird. 

Author Response

Thank you for your remark. I have changed the heading of the last section.